# Evaluation of the Bacterial Infections and Antibiotic Prescribing Practices in the Intensive Care Unit of a Clinical Hospital in Romania

**DOI:** 10.3390/antibiotics14010064

**Published:** 2025-01-09

**Authors:** Sándor Szabó, Bogdan Feier, Alina Mărginean, Andra-Elena Dumitrana, Simona Ligia Costin, Cecilia Cristea, Sorana D. Bolboacă

**Affiliations:** 1Department of Analytical Chemistry, Faculty of Pharmacy, “Iuliu Hațieganu” University of Medicine and Pharmacy, 4 Pasteur Street, 400349 Cluj-Napoca, Romania; szabo_sandor88@yahoo.com (S.S.); feier.george@umfcluj.ro (B.F.); 2“Dr. Constantin Papilian” Military Emergency Hospital, 400132 Cluj-Napoca, Romania; marginean.ali@gmail.com (A.M.); dumitrana_elena@yahoo.com (A.-E.D.); sincostin@yahoo.com (S.L.C.); 3Department of Medical Informatics and Biostatistics, Faculty of Medicine, “Iuliu Hațieganu” University of Medicine and Pharmacy Cluj-Napoca, 400349 Cluj-Napoca, Romania; sbolboaca@umfcluj.ro

**Keywords:** healthcare-associated infections (HAIs), bacteria, antibiotic (AB), intensive care unit (ICU), antibiotic resistance

## Abstract

**Introduction:** Healthcare-associated infections (HAIs) are associated with increased mortality, antimicrobial resistance, and high antibiotic use. **Methods**: The characteristics of bacterial resistance and antibiotic consumption in the intensive care unit (ICU) of a clinical hospital in Romania were evaluated. Demographic data of patients, identified bacteria, antibiotics administered, and their sensitivity profiles were collected and analyzed. **Results**: One hundred and twenty-five patients, with a median age of 68 years, mostly male (60%), were included in the study. More than one-third of the patients died. The deceased patients were older (median age of 74 years), had longer hospitalization (median of 9 days) and bacteria detected (55.3%), and had higher antibiotic consumption than the discharged patients. The most frequent bacteria identified in our cohort were *Acinetobacter baumannii*, *Klebsiella pneumoniae*, and *Pseudomonas aeruginosa* in deceased patients and *Klebsiella pneumoniae*, *Escherichia coli*, *Staphylococcus hemolyticus*, and *Enterococcus faecalis* in the survived group. The top three antibiotics used were ceftriaxone, metronidazole, and meropenem. Resistance to antibiotics was observed in 44.3% of the deceased group and 37.5% of patients who were discharged (χ^2^ = 5.5, *p* = 0.0628). **Discussion**: A positive monotonic association was observed between the number of hospitalization days and the number of antibiotic doses, with a higher correlation coefficient for deceased patients (0.6327, *p* < 0.0001) than in survived group (0.4749, *p* < 0.0001). **Conclusions and Future Trends**: This study provides a real picture of HAIs, the characteristics of bacteria, and the consumption of antibiotics in an ICU of a clinical hospital in Romania. The data obtained are similar to those from other international studies, but further studies are needed to reflect the real situation in Romania.

## 1. Introduction

Healthcare-associated infections (HAIs) are serious infections and a major public health problem [1,2] associated with a significant risk of morbidity, mortality, and prolonged hospital stay [3,4,5]. In Europe, approximately four million patients are infected annually in healthcare facilities [3,6]. Healthcare-associated infections are associated with a 4-fold higher mortality rate and a 3-fold longer hospitalization [7].

Admission to hospitals in emerging countries such as Romania is associated with a 15% risk of developing HAI [1,3], but HAIs remain under-reported [8]. The risk of occurrence of HAIs in developed countries is, on average, 5% (up to 16%); however, in the Intensive Care Unit (ICU), this risk can be higher and can increase to 30% compared to other hospital wards [3]. Intensive-care infections constitute approximately half of all HAIs [7]. The most common HAIs in the ICU are ventilator-associated pneumonia (VAP), urinary tract infections (UTIs), and bloodstream infections (BSIs) [1,4]. The presence of HAIs in patients hospitalized in the ICU prolongs hospitalization by five days [9]. HAIs are responsible for a direct cost of €7 billion annually [1]. HAIs increase the use of antibiotics, the emergence of multi-resistant bacteria, morbidity, mortality rates, and the cost of medical care [3,4,7]. The presence of HAIs is associated with higher antibiotic consumption as it requires the use of broad-spectrum antibiotics for a long period, thereby increasing the risk of antibiotic resistance [10]. For optimal antibiotic management, the implementation of antibiotic stewardship programs is needed. El-Sokkary et al. analyzed the data from 57 ICUs in 24 countries regarding infection prevention and control programs, and antibiotic stewardship activities and showed that most centers have a surveillance program [11]. Such programs may reduce the emergence of drug-resistant bacteria. In Romania, antibiotic-resistant infections account for over 60% of all HAI cases, compared to approximately 5% in Finland; however, the evaluated samples were small in some cases [1,12]. The most common multi-drug resistant (MDR) bacteria are defined using the acronym ESKAPE: *Enterococcus faecium*, *Staphylococcus aureus*, *Klebsiella pneumoniae*, *Acinetobacter baumannii*, *Pseudomonas aeruginosa*, and *Enterobacter* spp. [13]. Approximately 70% of patients in the ICUs receive antibiotic treatment [14]. The use of antimicrobials in European ICUs depends mainly on the availability of antibiotics; therefore, there are differences between countries in terms of antibiotic molecules used.

Romania has implemented measures to prevent and limit HAIs and to decrease antimicrobial resistance [15]; however, data on the bacteria and consumption of antibiotics in ICUs are limited. In this study, we aimed to investigate the etiology of bacterial infections in the ICU of a clinical hospital in Romania and to describe the bacterial pattern in collected biological samples and antibiotic consumption in deceased and surviving patients.

## 2. Results

### 2.1. The Evaluated Cohort

Four hundred and seventy-seven patients were hospitalized during the study period, of whom 125 met the inclusion criteria and were evaluated. More than one-third of the patients were included in the deceased group, and these patients had a longer hospitalization stay, and used a higher number of different antibiotics and antibiotic doses than those who survived (Table 1).

The top three biological samples collected were urine, bronchial secretions, and purulent/fluid secretions (Figure 1a), and the most frequent bacterium was *A. baumannii* (Figure 1b).

### 2.2. Types of Bacteria and Distribution in Collected Biological Samples

The most frequently identified bacteria were *A. baumannii*, *K. pneumoniae*, and *P. aeruginosa* in the deceased group and *K. pneumoniae*, *Escherichia coli*, *S. hemolyticus*, and *E. faecalis* in the surviving group (Figure 1b).

*K. pneumoniae* was the only bacteria identified in the cannula samples (two cases). One cerebral fluid culture revealed the presence of *K. pneumoniae*.

Considering the possibility of multiple biological samples being collected from patients, 41 positive cultures were identified in the deceased group and 51 were identified in the surviving group. Two bacteria with a prevalence higher than 10% were observed exclusively in the deceased group: *A. baumannii* and *K. pneumoniae*. The number of cases with positive cultures in biological samples by group are presented in Table 2.

### 2.3. Administered Antibiotics

Patients with no bacteria identified in their biological samples received up to three antibiotics (deceased vs. survived: 88:89% up to two antibiotics and 12:11% three antibiotics). Different antibiotics were administered, including antibiotics with the same or different pharmaceutical forms. The distribution of antibiotics administered to patients with multiple or no bacterial infections is presented in Table 3.

Table 4 summarizes the antibiotics used in deceased and surviving patients. The top five antibiotics used were ceftriaxone (34.2% deceased vs. 28.7% survived, *p* = 0.5405), metronidazole (15.8% deceased vs. 32.2% survived, *p* = 0.0581), meropenem (36.8% deceased vs. 20.7% survived, *p* = 0.0570), vancomycin (23.7% deceased vs. 11.5% survived, *p* = 0.0808), and amoxicillin/clavulanic acid (26.3% deceased vs. 8.0% survived, *p* = 0.0061).

The distribution of antibiotics by bacteria and group, considering that a patient could have more than one bacterium identified, and more than one antibiotic used, is presented in Table 5.

Specific bacteria were identified among deceased patients only:*Enterobacter cloacae* and prescribed antibiotics to patients: amoxicillin/clavulanic acid 1000/200 mg (*n* = 1), meropenem 1000 mg (*n* = 1), and vancomycin 1 g (*n* = 1).*Providencia stuartii* and prescribed antibiotics to patients: amoxicillin/clavulanic acid 1000/200 mg (*n* = 1), erythromycin 200 mg (*n* = 1), and vancomycin 1 g (*n* = 1).*Staphylococcus simulans* and the prescribed antibiotics were as follows: meropenem 1000 mg (*n* = 1), metronidazole 5 g/200 mL (*n* = 1), teicoplanin 400 mg (*n* = 1), trimethoprim/sulfamethoxazole 400/80 mg (*n* = 1), and vancomycin 1 g (*n* = 1).*Staphylococcus hominis* and prescribed antibiotics to patients: linezolid 2 mg/mL (*n* = 1), meropenem 1000 mg (*n* = 1), and vancomycin 1 g (*n* = 1).

Specific bacteria were exclusively observed in the surviving group:*Proteus penneri* and used antibiotics: ampicillin 1 g (*n* = 2), ceftriaxone 1 g (*n* = 1), colistin 1 MUI (*n* = 2), meropenem 1000 mg (*n* = 2), metronidazole 5 g/200 mL (*n* = 1), and vancomycin 1 g (*n* = 1).*Staphylococcus warneri* and used antibiotics: linezolid 2 mg/mL (*n* = 1) and meropenem 1000 mg (*n* = 1).

A positive monotonic association between hospitalization length and number of antibiotic doses used was found in the investigated cohort, with a higher correlation coefficient in deceased patients (ρ = 0.6327. *p* < 0.0001) than in the surviving group (ρ = 0.4749, *p* < 0.0001).

In the analyzed cohort, the hospitalization length was significantly different among patients with a different number of collected biological samples (from 0 to 5 samples, with five samples collected from two patients) (Kruskal–Wallis test: *p* < 0.0001, Figure 2). The difference remained statistically significant for deceased patients (Kruskal–Wallis test: *p* = 0.0124) only when patients without biological samples were compared to those with three biological samples (post hoc analysis, 0 vs. 3: *p* = 0.0055). In the surviving group, the number of days of hospitalization was also statistically significant (Kruskal–Wallis test: *p* = 0.0001), with the following significant differences in post hoc analysis: *p* = 0.0001 for 0 vs. 4, *p* = 0.0028 for 1 vs. 4, and *p* = 0.02621 for 2 vs. 4.

Statistically significant differences in hospitalization length for different numbers of bacteria identified in any biological sample were observed only in the survived group (Kruskal–Wallis test: *p* = 0.0046, post hoc analysis: 0 (n = 54) vs. 3 (n = 3), *p* = 0.0401) (Figure 3).

Statistically significant differences were observed between antibiotic doses for different numbers of bacteria identified in any biological sample only in survived patients (Kruskal–Wallis test: *p* = 0.0023, post hoc analysis: 0 (n = 54) vs. 2 (n = 3), *p* = 0.0114) (Figure 4b).

Statistically significant differences were observed between the antibiotic doses for different numbers of biological samples in deceased patients (Kruskal–Wallis test: *p* = 0.0132, post hoc analysis: 0 (n = 8) vs. 2 (n = 7), *p*= 0.0136), as well as in those who survived (Kruskal–Wallis test: *p* = 0.0005, post hoc analysis: 0 (n = 35) vs. 4 (n = 7), *p* = 0.0261) (Figure 5).

Significant correlations were observed between the number of identified bacteria and the number of biological samples, with a similar value for patients in the deceased (Kendall tau correlation coefficient = 0.569, *p* < 0.001) and survived group (Kendall tau correlation coefficient = 0.628, *p* < 0.001).

### 2.4. Antibiotic Resistance Profile

More than 50% of the identified bacteria were resistant to antibiotics. Resistance to antibiotics was observed in 44.3% of the deceased patients and in 37.5% in the survived patients (χ^2^ = 5.5, *p* = 0.0628). Four hundred and thirty-two resistance profiles were identified for the evaluated antibiotics and the top three antibiotics were ciprofloxacin (42 cases), gentamicin (41 cases) and TMP/SMTX (36 cases) (Figure 6).

In our cohort, 631 bacterial strains were susceptible to antibiotics. The top antibiotic sensitivity was found in 40 patients (gentamicin), 37 patients (linezolid), and 35 patients (ciprofloxacin, fosfomycin, tigecycline, and TMP/SMTX). In the opposite pole, the smallest number of sensible cases were (a) ticarcillin (one case), (b) aztreonam and oxacillin (three cases each), and (c) erythromycin, rifampicin, and tetracycline (seven cases each). In our cohort, the identified bacteria were sensitive to half of the investigated antibiotics in up to 26 cases.

## 3. Discussion

*A. baumannii*, *K. pneumoniae*, and *P. aeruginosa* were the most frequently observed bacteria among deceased patients, whereas *K. pneumoniae*, *E. coli*, *S. hemolyticus*, and *E. faecalis* were most frequently observed among those who survived. Statistically significant differences were observed between the deceased patients and the surviving group in terms of length of hospitalization, number of bacteria, number of antibiotics, and total number of antibiotic doses. The bacteria with a higher number of resistances identified in our cohort were *K. pneumoniae*, *S. hemolyticus*, and *A. baumannii.* To the best of our knowledge, this is the first study in Romania that provides a report of bacterial infections in an ICU, antibiotic resistance, and antibiotic consumption. The results of our study provide information on the characteristics of bacteria identified in ICU patients regarding their resistance or sensitivity to antibiotics and highlight antibiotic prescription practices in terms of classes and number of antibiotic doses used.

Half of the investigated patients were older than 68 years, underlining the increase in life expectancy in Romania; therefore, critical patients are getting older. The deceased patients in our cohort were older, but the significance threshold was not reached, indicating fragile health and a higher death rate compared to patients who survived.

More than a third of the patients in the evaluated cohort did not survive, and those who died had longer hospitalization lengths (on average three days longer) and a higher number of different antibiotics and antibiotic doses than those who survived. Our results are in line with those reported in the scientific literature; longer stays in the ICU increase the risk of developing HAIs, hospitalization costs, and mortality [16].

Samples (most commonly, nasal, rectal, and skinfold portage) are taken when a patient is admitted to the ICU to screen for bacterial colonization and MDR bacteria. Blood cultures are performed when septicemia is suspected or when the source of infection cannot be identified. In our study, the top three collected biological samples were urine, bronchial secretions, and purulent/fluid secretions, reflecting the particularities of the investigated hospital.

Most bacteria involved in HAIs are of endogenous origin, but they can also be acquired from human or environmental sources [17]. The most common bacteria involved in HAIs are Gram-negative bacteria, with *E. coli* and *P. aeruginosa* as the most frequently isolated agents. The most frequent bacteria are *Klebsiella* spp. in ICU-acquired pneumonia, coagulase-negative staphylococci in ICU-acquired BSIs, and *E. coli* in ICU-acquired UTIs [12]. Septic episodes in the ICU are often caused by MDR pathogens.

Some bacteria are constantly found in the ICU [18], and in this study, we found that the three most frequent bacteria were *A. baumannii*, *K. pneumoniae*, and *P. aeruginosa* in the deceased group and *K. pneumoniae*, *E. coli*, *S. hemolyticus*, and *E. faecalis* in the survived group. In the deceased group, *A. baumannii* and *K. pneumoniae* were present, with a prevalence of >10%. These data suggest that some bacteria are constantly present in the ICU and contribute to the occurrence of HAIs. We noticed that some patients had multiple infections, with several bacteria identified from the collected biological samples, similar to the results reported by Barbato et al. [19]. The presence of multiple infections has led to the use of several antibiotics, especially if the bacteria have different sensitivities to the tested antibiotics. The top three bacteria identified in our cohort were *A. baumannii* (most frequent), *K. pneumoniae* and *P. aeruginosa* (second place), *E. faecalis* and *E. coli* (third position). *A. baumannii* can survive for prolonged periods in hospital environments and on medical devices and can spread from one person to another through contact with contaminated surfaces or equipment or through person-to-person transmission, often via contaminated hands [20]. *K. pneumoniae* is a common pathogen isolated in the ICU, seen less frequently in patients admitted directly from the community (6%, 95% confidence interval [CI] [3% to 8%]) than in those with recent medical care (19%, 95% CI [14% to 51%]) [21]. *P. aeruginosa*, a Gram-negative bacterium frequently identified in HAIs, occurs at a frequency of 11.6%, according to Harris et al. [22], a result similar to our findings. *E. faecalis*, a low-virulence bacterium, has been reported in ICUs with a frequency of less than 20%, 17% (498,998 analyzed patients with 1,554,070 patient-days, United States from 1992 to 1998) [23], and 14.8% (blood culture isolates reported in a sample of 1327 ICU patients in India) [24]. *E. coli* bacteremia is associated with a mortality rate of 18.2% [95% CI (17.8 to 18.7%)] in England, based on 28,616 samples from a national study [25]. The frequencies of the listed bacteria identified in our sample were different from those reported in the scientific literature, with a higher predominance of *A. baumannii*.

One of the inclusion criteria in our study was the use of at least one antibiotic in hospitalized patients, with almost 40% of deceased patients receiving more than two antibiotics, and the same percentage of discharged patients receiving two antibiotics. Some patients with negative cultures received up to three different antibiotics, which could be explained by using antibiotics as empirical treatment until the culture results were obtained. A tendency to overuse antibiotics was observed, and multiple factors contributed to this tendency.

Many factors influence the choice of antibiotic therapy, among which, patient characteristics, suspected site of infection, local microbial susceptibility, and antibiotic stewardship programs are the most important. The cost, availability of antibiotics, and allergies are also considered [26]. Except for patients with septic shock, the following guidelines were applied: refraining from immediate antibiotic prescription, limiting broad-spectrum antibiotics to patients without risk factors for MDR organisms, switching to monotherapy as soon as possible, and narrowing the spectrum when cultures and susceptibility results were available. However, in the ICU, the decision to prescribe broad-spectrum antibiotics is context-dependent, guided by the fear of legal and ethical consequences in cases of insufficient treatment, and the toxic impact on the individual and society is less often considered [27]. Thus, the ICU is at the top of antibiotic consumption and is the headquarters for the development and selection of MDR bacteria.

Patients with several strains of bacteria usually receive combined antibiotic treatment. Whenever ineffective, antibiotics are changed and, thus, during the same hospitalization, the patient may receive several classes of antibiotics. As post-surgical patients are also transferred to the ICU, metronidazole consumption is higher to cover the anaerobic flora.

Therapeutic plans often suggest a combination of antibiotics to treat multiple bacterial infections, including those with potential resistance to antibiotics. In addition, a challenge for the medical staff in the ICU is the fact that most of the time, the patients are in critical condition, they can have organ dysfunctions and other associated diseases leading to changes in the pharmacokinetics of antibiotics [28], thus increasing the risk of over or underdosing of antibiotics with unpredictable therapeutic outcomes [29]. To adapt the antibiotic doses to the patient in the ICU, an effective collaboration is needed between the attending physician, infectious disease doctor/epidemiologist, and clinical pharmacist following antimicrobial and diagnostic stewardship principles [30].

As expected, there was a direct correlation between the number of identified bacteria and the number of antibiotics from different classes administered to the patients. However, in our cohort, more than 55% received one to three antibiotics, although the bacterial culture results were not known, with empirical administration, like other European data [31]. During hospitalization, some patients received one to three antibiotics, although only one bacterium was identified. Other cases have been identified in which infections with multiple bacteria were treated with up to seven different antibiotics during hospitalization. The top three most frequently used antibiotics were ceftriaxone, metronidazole, and meropenem, which were administered to over 30% of patients. Ceftriaxone was widely used in the evaluated cohort mainly because it was accessible, cheap, and included in different treatment protocols. As per current practice, most surgical patients are transferred to the ICU for several days. Some surgical interventions involve the risk of infection with anaerobic bacteria, that explains the use of metronidazole. The local resistance to meropenem in the analyzed ICU is low; therefore, meropenem is often recommended as the first chosen antibiotic.

Ghiga et al. [32] underline the fact that over 3% of the population in Romania uses antibiotics every day, with over a million doses of antibiotics consumed daily. Romania ranked first in 2021 among the European Union countries in terms of antibiotic consumption, which can explain the existence of high antimicrobial resistance. Data reported in 2020 from 16 hospitals in Romania highlighted the methicillin-resistant *S. aureus* (MRSA) strains representing 47.4% of all reported *S. aureus* strains, and glycopeptide resistance of *E. faecium* (Vancomycin-resistant Enterococci) increased to 41.7% from 2.9% in eight years (2012–2020). Resistance to carbapenems was 93.6% for *A. baumannii*, 89.4% for MDR, and 46.5 % for *K. pneumoniae*. However, resistance to aminoglycosides and cephalosporins has been reduced [33].

Resistance to third-generation cephalosporins has been reported in the scientific literature in 15% of *E. coli* isolates, 38% of *Klebsiella* spp., and 37% of *Enterobacter* spp. isolates [4,12]. Some strains of bacteria show multiple antibiotic resistances; thus, multiple classes of antibiotics were used from the tested cohort for the same strain of bacteria. In the deceased group, 47.4% had at least one resistant bacterium compared with the discharged patients (32.2%). The highest antibiotic resistance was observed in *K. pneumoniae*, *S. hemolyticus*, and *A. baumannii*, whereas the top three antibiotic resistances were identified for ciprofloxacin, gentamicin, and TMP/SMTX in our cohort.

The final susceptibility testing results are available two days after biological sample collection, and once the results are ready, the antibiotic treatment may be changed. The prescription date of antibiotics is not always identical to the first day of treatment because the antibiotics available in the ICU are used and reported in the patient’s electronic file only after a few days. Antibiotic therapy can be initiated in a different hospital ward and continued in the ICU, resulting in fewer doses registered electronically in the ICU than the length of stay. Furthermore, the lack of antibiotics in the hospital pharmacy may lead to non-compliance with susceptibility test results.

In cases where the patient dies on the day when the medication is prescribed, the entire dose of antibiotic prescribed remains recorded as the administered medication, which is a lack of the hospital’s electronic health record system. All the limitations that were identified should be taken into consideration by the hospital management and authorities who are struggling to reduce antibiotic consumption and could be implemented in the hospital guidelines.

Because classic methods for identifying bacteria in biological samples are time-consuming, new methods have been developed to reduce the time required to obtain the results. A series of rapid methods for detecting bacteria in biological samples have been studied, and some have been applied in hospitals. Analytical techniques (LC-MS, fluorescence-based methods, MALDI-TOF), immunological techniques (lateral flow immunoassay), and molecular techniques (ELISA, PCR, LAMP) have multiple advantages, such as good sensitivity, but they also have drawbacks, such as cost per analysis and the need for trained personnel [34,35]. In this context, electrochemical and optical (bio)sensors are fast, easy to use, and sensitive alternatives, capable of detecting whole bacteria or molecules secreted by the bacteria (e.g., virulence factors or quorum sensing molecules) [36].

To obtain rapid access to information related to the presence of pathogenic bacteria or for the monitoring of antibiotics from biological samples, electrochemical and optical sensors embedded in point-of-care (POC) devices have been developed. Rapid diagnosis of bacteria is important in the management of HAIs and AMR with multilevel implications: prevention strategies, rapid diagnosis to enable personalized management of therapies, and follow-ups to remove pressure from healthcare units and diminish the economic burden. The literature mentions sensing devices based on various techniques for bacterial identification; however, they cannot be considered as POC yet but as preliminary devices in their development [34].

## 4. Methods

### 4.1. Participants and Data Collection

A retrospective evaluation of bacterial infections and antibiotic use was conducted in the ICU of a clinical hospital in Cluj-Napoca between 1 February 2020 and 31 December 2020. This 11-month period was chosen because from the 1st of January 2021, the ICU became a COVID–ICU ward, and the therapeutic approach for these patients was not part of the objective of the current study.

The electronic medical health record system was used as the source of raw data. Patients with a negative COVID test result and no treatment associated with COVID who were hospitalized for at least three days in the ICU and received at least one antibiotic were included in the study.

### 4.2. Identification of Bacteria

Biological samples were collected from patients before receiving antimicrobial therapy. Biological samples were sent to the hospital’s laboratory no later than 3:00 p.m. and processed using Petri dishes for bacterial cultures. The next day, the results were interpreted by a laboratory physician, cultures were identified, and susceptibility tests were carried out. The results were subsequently validated. These steps last approximately 48 h. Although the bacterial identification method is fast, obtaining a susceptibility test leads to an extension of time.

Bacteria were routinely identified using morphological and biochemical tests. Gram-negative and Gram-positive bacteria were identified using a VITEK 2 Systems analyzer (bioMerieux, Marcy-l’Étoile, France) for the automated identification of the most clinically significant fermenting and non-fermenting Gram-negative bacilli. The results are available in approximately 10 h or less [37]. Rapid susceptibility results were obtained from 3.3 to 17.5 h [37]. The results for Gram-positive bacteria are available in approximately 8 h or less [32]. Antimicrobial susceptibility test results regarding rapid susceptibility are obtained in 6 to 17 h [33]. VITEK 2 uses computer-assisted analysis of growth on plastic cards to calculate the minimum inhibitory concentration (MIC).

### 4.3. Statistical Analysis

The data were summarized according to the primary outcome. Descriptive statistics metrics were reported as follows: number and percentages for attribute date, median [Q1 to Q3] (where Q1 is the 25th percentile and Q3 is the 75th percentile) for quantitative discrete and continuous variables. Differences between groups were tested using the Chi-squared or Fisher’s exact test according to the expected frequencies of the attribute data. The Mann–Whitney U test or Kruskal–Wallis test was used to compare quantitative data between two or more groups. Whenever the Kruskal–Wallis test showed statistically significant differences between groups, a post hoc analysis was conducted. The Kendall tau correlation coefficient was calculated to quantify the association between the number of identified bacteria and biological samples.

Bacterial cultures were performed using different biological samples (e.g., cannula, cerebrospinal fluid, blood, urine, bronchial secretion, and purulent/fluid secretion). In some cases, more than one biological sample was collected from the same. In addition, more than one bacterium was identified in biological samples from the same patient. Antibiotic susceptibility tests were performed for any bacteria identified in the biological samples. Therefore, the total number of identified bacteria was higher than that of patients in the analyzed cohort.

Exploratory statistical analysis was conducted using Statistica software (v.13.5, TIBCO Software Inc., Palo Alto, CA, USA). Graphical representations were created using Microsoft Excel (Microsoft Office 365, USA) or JASP (v. 0.18.3.0). All tests were two-tailed at a significance level of 5%, and *p*-values less than 0.05 were considered statistically significant.

## 5. Conclusions and Future Trends

Infection with *K. pneumoniae* was observed in the cohort, regardless of the group, and with *A. baumannii* and *P. aeruginosa* among deceased patients, and *E. coli*, *S. hemolyticus*, and *E. faecalis* among discharged patients. The deceased group was older and had a longer hospitalization stay, higher bacterial load, number of antibiotics used, and higher total antibiotic doses administered. Bacteria with the highest levels of resistance to antibiotics identified in this cohort were *K. pneumoniae*, *S. hemolyticus*, and *A. baumannii*. Our results highlighted the main challenges recorded in one ICU of a clinical hospital in Romania in terms of HAIs, identified bacteria, and the consumption of antibiotics. Our results could be used to develop guidelines for healthcare professionals and support measures for more judicious use of antibiotics. Rapid detection methods for bacteria, as well as careful and continuous antibiotic surveillance, not only in the ICU, are needed to guide healthcare workers to provide the best patient care.

## Figures and Tables

**Figure 1 antibiotics-14-00064-f001:**
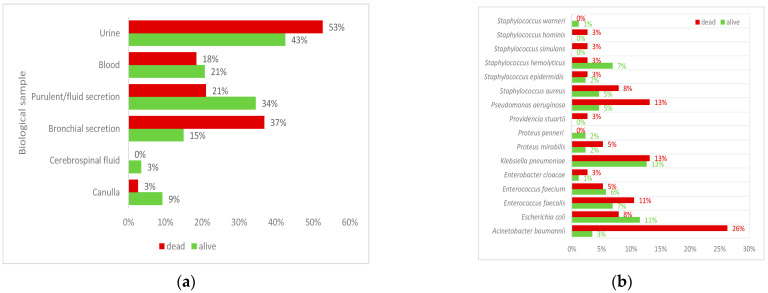
(**a**) Distribution of collected biological samples (38 deceased patients and 87 who survived); (**b**) distribution of bacteria per group expressed as the number of positive cases per number of eligible cases, considering that the same patient can have more than one bacterium).

**Figure 2 antibiotics-14-00064-f002:**
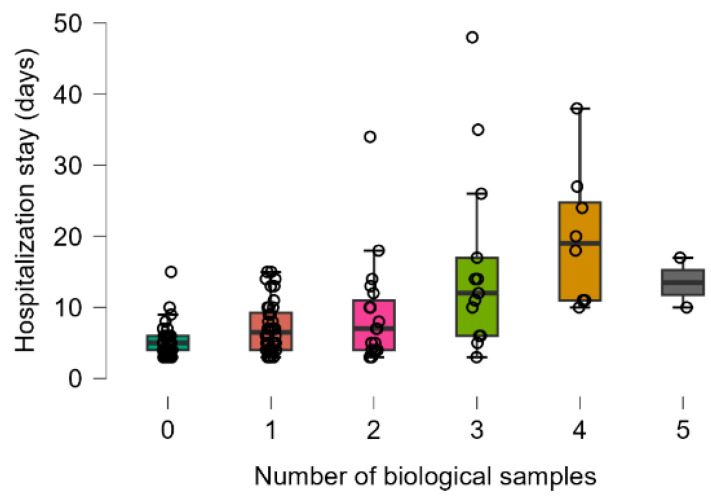
Distribution of number of hospitalization days by number of biological samples (post hoc analysis: 0 vs. 3 *p* = 0.0007, 0 vs. 4 *p* < 0.0001, 1 vs. 4 *p* = 0.0069). Dots represent the raw data, the box middle line is the median, the boxes are the first, and the third quartile and the wickers are the minimum and maximum (excluding the outliers). Colors represent the distribution of the length of hospital stay for each number of biological samples.

**Figure 3 antibiotics-14-00064-f003:**
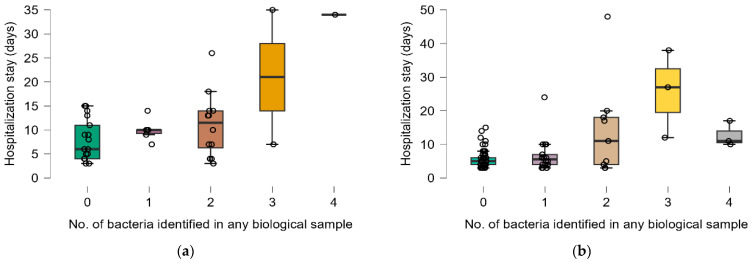
Distribution of hospitalization length and number of identified bacteria by groups: (**a**) deceased, (**b**) survived. Dots represent the raw data, the box middle line is the median, the boxes are the first, and the third quartile and the wickers are the minimum and maximum (excluding the outliers). Colors represent the distribution of the length of hospital stay for each number of identified bacteria.

**Figure 4 antibiotics-14-00064-f004:**
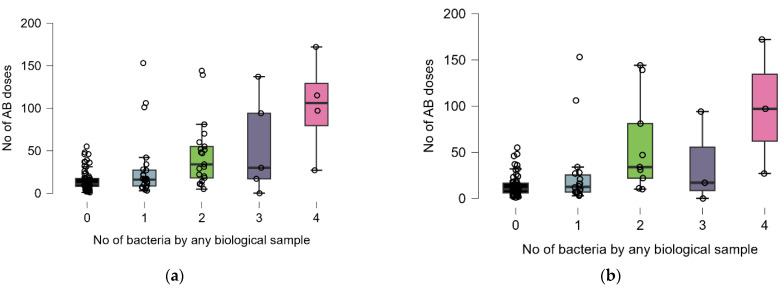
Distribution of number of antibiotic doses and number of identified bacteria by groups: (**a**) deceased and (**b**) survived. Dots represent the raw data, the box middle line is the median, the boxes are the first, and the third quartile and the wickers are the minimum and maximum (excluding the outliers). Colors represent the distribution of antibiotic doses for each number of biological sample.

**Figure 5 antibiotics-14-00064-f005:**
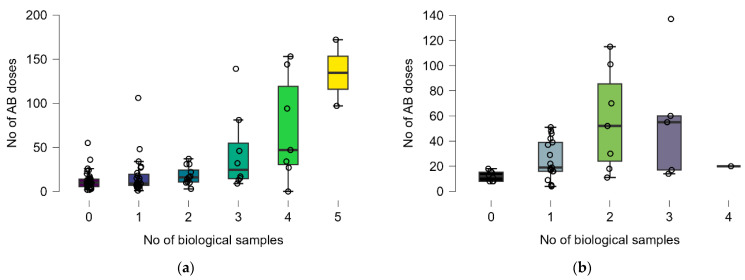
Distribution of number of antibiotics doses and number of biological samples by groups: (**a**) deceased and (**b**) survived. Dots represent the raw data, the box middle line is the median, the boxes are the first, and the third quartile and the wickers are the minimum and maximum (excluding the outliers). Colors represent the distribution of antibiotic doses for each number of biological sample.

**Figure 6 antibiotics-14-00064-f006:**
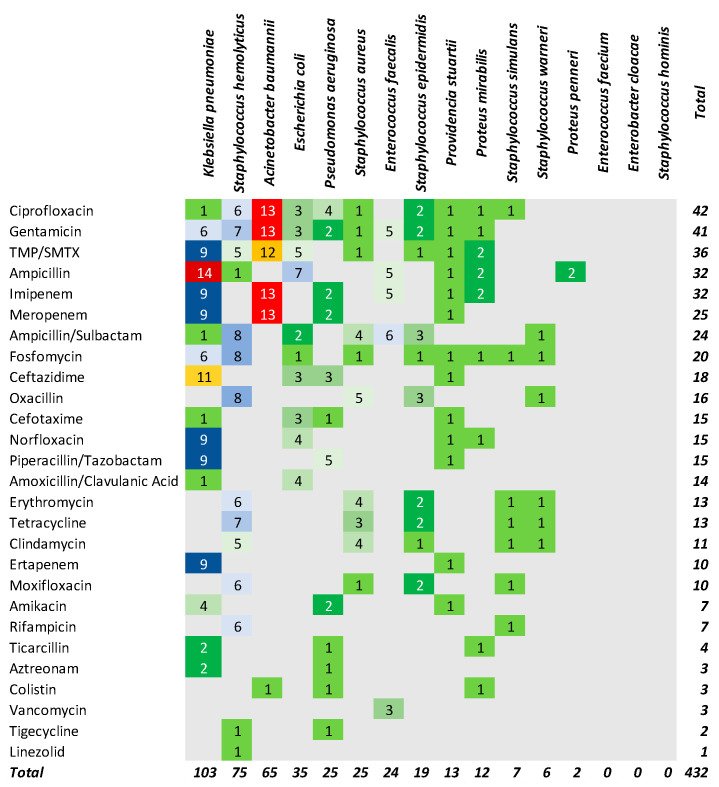
The matrix of bacteria by antibiotic resistance. A specific color was used for a specific number of antibiotic resistances. Grey background indicates the absence of resistance.

**Table 1 antibiotics-14-00064-t001:** Characteristics of the evaluated cohort.

Characteristic	All (*n* = 125)	Deceased (*n* = 38)	Survived (*n* = 87)	*p*-Value
Age ^a^				0.1794
Median [Q1 to Q3]	68 [62 to 79]	74 [63.3 to 81.8]	68 [61 to 77.5]	
{min to max}	{39 to 94}	{48 to 93}	{39 to 94}	
Sex ^b^				0.3825
Women	50 (40)	13 (34.2)	37 (42.5)	
Men	75 (60)	25 (65.8)	50 (57.5)	
Days of hospital stay ^a^				0.0031
Median [Q1 to Q3]	6 [4 to 10]	9 [5.3 to 13.8]	6 [4 to 8]	
{min to max}	{3 to 48}	{3 to 35}	{3 to 48}	
No. of biological samples ^b^				0.1036
0	43 (34.4)	8 (21.1)	35 (40.2)	
1	40 (32)	17 (44.7)	23 (26.4)	
2	19 (15.2)	7 (18.4)	12 (13.8)	
>2	23 (18.4)	6 (15.8)	17 (19.5)	
No. of bacteria ^b,^*				0.0342
0	71 (56.8)	17 (44.7)	54 (62.1)	
1	24 (19.2)	6 (15.8)	18 (20.7)	
2	21 (16.8)	12 (31.6)	9 (10.3)	
>2	9 (7.2)	3 (7.9)	6 (6.9)	
No. of antibiotics ^b^				0.0261
1	47 (37.9)	13 (34.2)	34 (39.5)	
2	44 (35.5)	9 (23.7)	35 (40.7)	
>2	33 (26.6)	16 (42.1)	17 (19.8)	
No. of patients with at least one ^c^				
R	46 (36.8)	18 (47.4)	28 (32.2)	0.1054
I	17 (13.6)	8 (21.1)	9 (10.3)	0.1082
S	53 (42.4)	2330 (52.6)	33 (37.9)	0.1261
Total no. of antibiotic doses ^a^				0.0040
Median [Q1 to Q3]	15 [9 to 31]	18 [14.3 to 45]	13 [7.5 to 26.5]	
{min to max}	{0 to 172}	{4 to 137}	{0 to 172}	

^a^ median [Q1 to Q3], {min to max}, where Q1 is the first quartile, Q3 is the third quartile, min is then minimum, max is the maximum; comparison between groups with Mann–Whitney test. ^b^ no (%), comparison with Chi-squared test or * Fisher’s exact test. ^c^ Number of cases with at least one case of R = resistance; I = intermediary; S = sensible.

**Table 2 antibiotics-14-00064-t002:** Distribution of bacteria in biological samples by group.

Bacterium	Biological Sample	No. Patients (Percentage, %)		
		All, *n* = 125	Deceased, *n* = 38	Survived, *n* = 87
*Acinetobacter baumannii*	Bronchial secretion	9 (7.2)	7 (18.4)	2 (2.3)
	Purulent/fluid secretion	4 (3.2)	3 (7.9)	1 (1.1)
*Enterobacter cloacae*	Bronchial secretion	2 (1.6)	1 (2.6)	1 (1.1)
*Enterococcus faecalis*	Blood	2 (1.6)	1 (2.6)	1 (1.1)
	Urine	2 (1.6)	1 (2.6)	1 (1.1)
	Purulent/fluid secretion	5 (4)	2 (5.3)	3 (3.4)
*Enterococcus faecium*	Blood	1 (0.8)	1 (2.6)	
	Urine	3 (2.4)	1 (2.6)	2 (2.3)
	Purulent/fluid secretion	4 (3.2)	1 (2.6)	3 (3.4)
*Escherichia coli*	Blood	1 (0.8)		1 (1.1)
	Urine	4 (3.2)		4 (4.6)
	Bronchial secretion	1 (0.8)		1 (1.1)
	Purulent/fluid secretion	7 (5.6)	3 (7.9)	4 (4.6)
*Klebsiella pneumoniae*	Cannula	2 (1.6)		2 (2.3)
	Cerebrospinal fluid	1 (0.8)		1 (2.6)
	Blood	1 (0.8)		1 (1.1)
	Urine	2 (1.6)		2 (2.3)
	Bronchial secretion	9 (7.2)	5 (13.2)	4 (4.6)
	Purulent/fluid secretion	3 (2.4)		3 (3.4)
*Proteus mirabilis*	Urine	2 (1.6)		2 (2.3)
	Bronchial secretion	1 (0.8)	1 (2.6)	
	Purulent/fluid secretion	1 (0.8)	1 (2.6)	
*Proteus penneri*	Purulent/fluid secretion	2 (1.6)		2 (2.3)
*Providencia stuarti*	Blood	1 (0.8)	1 (2.6)	
*Pseudomonas aeruginosa*	Urine	3 (2.4)	3 (7.9)	
	Bronchial secretion	2 (1.6)	2 (5.3)	
*Staphylococcus aureus*	Bronchial secretion	2 (1.6)	1 (2.6)	2 (2.3)
	Purulent/fluid secretion	5 (4)	2 (5.3)	1 (1.1)
	Urine	1 (0.8)		1 (1.1)
*Staphylococcus epidermidis*	Urine	1 (0.8)	1 (2.6)	
	Bronchial secretion	2 (1.6)		2 (2.3)
	Blood	1 (0.8)		1 (1.1)
*Staphylococcus hemolyticus*	Blood	4 (3.2)	1 (2.6)	3 (3.4)
	Urine	1 (0.8)		1 (1.1)
	Purulent/fluid secretion	2 (1.6)		2 (2.3)
*Staphylococcus hominis*	Blood	1 (0.8)	1 (2.6)	
*Staphylococcus simulans*	Blood	1 (0.8)	1 (2.6)	
*Staphylococcus warneri*	Purulent/fluid secretion	1 (0.8)		1 (1.1)

Data are expressed as number and corresponding percentage (%).

**Table 3 antibiotics-14-00064-t003:** The number of antibiotics administered per number of identified bacteria stratified by group.

No. of Antibiotics for	No. Patients (Percentage, %)		
	All, *n* = 125	Deceased, *n* = 38	Survived, *n* = 87
*no bacteria*	*71 (56.8)*	*17 (44.7)*	*54 (62.1)*
1 AB	32 (45.1)	8 (47.1)	24 (44.4)
2 ABs	31 (43.7)	7 (41.2)	24 (44.4)
3 ABs	8 (11.3)	2 (11.8)	6 (11.1)
*1 bacterium*	*24 (19.2)*	*6 (15.8)*	*18 (20.7)*
1 AB	10 (41.7)	2 (33.3)	8 (44.4)
2 ABs	8 (33.3)	1 (16.7)	7 (38.9)
3 ABs	6 (25)	3 (50)	3 (16.7)
*2 bacteria*	*21 (16.8)*	*12 (31.6)*	*9 (10.3)*
1 AB	5 (23.8)	3 (25)	2 (22.2)
2 ABs	3 (14.3)	1 (8.3)	2 (22.2)
3 ABs	8 (38.1)	6 (50)	2 (22.2)
>3 ABs	5 (23.8)	2 (16.7)	3 (33.3)
*3 bacteria*	*5 (4)*	*2 (5.3)*	*3 (3.4)*
no AB	1 (20)	0 (0)	1 (33.3)
2 ABs	1 (20)	0 (0)	1 (33.3)
3 ABs	1 (20)	1 (50)	0 (0)
>3 ABs	2 (40)	1 (50)	1 (33.3)
4 bacteria	*4 (3.2)*	*1 (2.6)*	*3 (3.4)*
2 ABs	1 (25)	0 (0)	1 (33.3)
3 ABs	1 (25)	0 (0)	1 (33.3)
>3 ABs	2 (50)	1 (100)	1 (33.3)

Data are reported as no (%). AB = antibiotic.

**Table 4 antibiotics-14-00064-t004:** The use of antibiotics by group.

Antibiotic	No. Patients (Percentage, %)		
	All, *n* = 125	Deceased, *n* = 38	Survived, *n* = 87
Amikacin 500 mg	3 (2.4)	1 (2.6)	2 (2)
Amoxicillin/Clavulanic acid 1000/200 mg	17 (13.6)	10 (26.3)	7 (7.1)
Ampicillin 1 g	3 (2.4)	1 (2.6)	2 (2)
Azithromycin 500 mg	2 (1.6)	1 (2.6)	1 (1)
Cefazolin 1 g	1 (0.8)	0 (0)	1 (1)
Cefoperazon/Sulbactam 1000/1000 mg	20 (16)	2 (5.3)	18 (18.4)
Ceftazidime 1 g	3 (2.4)	2 (5.3)	1 (1)
Ceftriaxone 1 g	38 (30.4)	13 (34.2)	25 (25.5)
Cefuroxime 1.5 g	15 (12)	0 (0)	15 (15.3)
Clindamycin 300 mg/2 mL	4 (3.2)	1 (2.6)	3 (3.1)
Colistin 1MUI	13 (10.4)	6 (15.8)	7 (7.1)
Doxycycline 100 mg	4 (3.2)	2 (5.3)	2 (2)
Erythromycin 200 mg	4 (3.2)	1 (2.6)	3 (3.1)
Ertapenem 1 g	7 (5.6)	1 (2.6)	6 (6.1)
Fosfomycin 3 g p.o.	1 (0.8)	0 (0)	1 (1)
Gentamicin 80 mg	2 (1.6)	0 (0)	2 (2)
Linezolid 2 mg/mL	2 (1.6)	1 (2.6)	1 (1)
Meropenem 1000 mg	32 (25.6)	14 (36.8)	18 (18.4)
Metronidazole 250 mg	10 (8)	2 (5.3)	8 (8.2)
Metronidazole 5 g/200 mL	34 (27.2)	6 (15.8)	28 (28.6)
Moxifloxacin 400 mg/250 mL	2 (1.6)	1 (2.6)	1 (1)
Oxacillin 1000 mg	2 (1.6)	2 (5.3)	0 (0)
Penicillin G Potassium1 MUI	1 (0.8)	0 (0)	1 (1)
Piperacillin/Tazobactam 2 g/0.25 g	6 (4.8)	4 (10.5)	2 (2)
Rifampin 300 mg caps.	1 (0.8)	1 (2.6)	0 (0)
Tigecycline 50 mg	2 (1.6)	0 (0)	2 (2)
Teicoplanin 400 mg	1 (0.8)	1 (2.6)	0 (0)
Trimethoprim/Sulfamethoxazole 400/80 mg	4 (3.2)	3 (7.9)	1 (1)
Vancomycin 1 g	19 (15.2)	9 (23.7)	10 (10.2)

Data reported as a number (%).

**Table 5 antibiotics-14-00064-t005:** Distribution of administered antibiotics by bacteria and group.

Bacterium	Administered Antibiotic	
	Deceased, *n* = 38	Survived, *n* = 87
*Acinetobacter baumannii*	Amikacin 500 mg (*n* = 1) Amoxicillin/Clavulanic acid 1000/200 mg (*n* = 1) Ampicillin 1 g (*n* = 1) Ceftazidime 1 g (*n* = 2) Ceftriaxone 1 g (*n* = 3) Colistin 1 MUI (*n* = 6) Ertapenem 1 g (*n* = 1) Meropenem 1000 mg (*n* = 6) Metronidazole 5 g/200 mL (*n* = 3) Piperacillin/Tazobactam 2 g/0.25 g (*n* = 2) Rifampin 300 mg caps. (*n* = 1) Trimethoprim/Sulfamethoxazole 400/80 mg (*n* = 2) Vancomicyn 1 g (*n* = 3)	Ceftriaxone 1 g (*n* = 1) Colistin 1 MUI (*n* = 3) Erythromicin 200 mg (*n* = 1) Meropenem 1000 mg (*n* = 3) Metronidazole 5 g/200 mL (*n* = 1) Piperacillin/Tazobactam 2 g/0.25 g (*n* = 1) Vancomycin 1 g (*n* = 2)
*Escherichia coli*	Colistin 1 MUI (*n* = 1) Ertapenem 1 g (*n* = 1) Meropenem 1000 mg (*n* = 3) Metronidazole 250 mg (*n* = 1) Metronidazole 5 g/200 mL (*n* = 1) Rifampin 300 mg caps. (*n* = 1) Vancomycin 1 g (*n* = 1)	Ampicillin 1 g (*n* = 2) Cefazolin 1 g (*n* = 1) Cefoperazone/Sulbactam 1000/1000 mg (*n* = 1) Ceftriaxone 1 g (*n* = 3) Colistin 1 MUI (*n* = 2) Ertapenem 1 g (*n* = 3) Meropenem 1000 mg (*n* = 3) Metronidazole 250 mg (*n* = 1) Metronidazole 5 g/200 mL (*n* = 5) Vancomycin 1 g (*n* = 1)
*Enterococcus faecalis*	Amikacin 500 mg (*n* = 1) Colistin 1 MUI (*n* = 2) Meropenem 1000 mg (*n* = 4) Metronidazole 5 g/200 mL (*n* = 1) Teicoplanin 400 mg (*n* = 1) Trimethoprim/Sulfamethoxazole 400/80 mg (*n* = 2) Vancomycin 1 g (*n* = 1)	Cefazolin 1 g (*n* = 1) Ceftriaxone 1 g (*n* = 4) Doxycycline 100 mg (*n* = 1) Linezolid 2 mg/mL (*n* = 1) Meropenem 1000 mg (*n* = a) Metronidazole 5 g/200 mL (*n* = 2)
*Enterococcus faecium*	Ceftriaxone 1 g (*n* = 2) Colistin 1 MUI (*n* = 1) Meropenem 1000 mg (*n* = 1) Vancomycin 1 g (*n* = 1)	Ampicillin 1 g (*n* = 2) Ceftriaxone 1 g (*n* = 3) Colistin 1 MUI (*n* = 2) Erythromycin 200 mg (*n* = 1) Ertapenem 1 g (*n* = 1) Meropenem 1000 mg (*n* = 2) Metronidazole 5 g/200 mL (*n* = 2) Vancomycin 1 g (*n* = 1)
*Klebsiella pneumoniae*	Amoxicillin/Clavulanic acid 1000/200 mg (*n* = 2) Ampicillin 1 g (*n* = 1) Ceftazidime 1 g (*n* = 2) Ceftriaxone 1 g (*n* = 1) Colistin 1 MUI (*n* = 2) Meropenem 1000 mg (*n* = 1) Metronidazole 5 g/200 mL (*n* = 1) Piperacillin/Tazobactam 2 g/0.25 g (*n* = 2) Vancomycin 1 g (*n* = 1)	Amikacin 500 mg (*n* = 1) Amoxicillin/Clavulanic acid 1000/200 mg (*n* = 1) Ampicillin 1 g (*n* = 2) Cefoperazone/Sulbactam 1000/1000 mg (*n* = 1) Ceftriaxone 1 g (*n* = 3) Clindamycin 300 mg/2 mL (*n* = 1) Colistin 1 MUI (*n* = 4) Ertapenem 1 g (*n* = 2) Fosfomycina 3 g p.o. (*n* = 1) Meropenem 1000 mg (*n* = 8) Metronidazole 250 mg (*n* = 2) Metronidazole 5 g/200 mL (*n* = 2) Tigecycline 50 mg (*n* = 2) Vancomycin 1 g (*n* = 2)
*Proteus mirabilis*	Amoxicillin/Clavulanic acid 1000/200 mg (*n* = 1) Meropenem 1000 mg (*n* = 1) Oxacillin 1000 mg (*n* = 1) Trimethoprim/Sulfamethoxazole 400/80 mg (*n* = 1)	Cefoperazone/Sulbactam 1000/1000 mg (*n* = 1) Ceftriaxone 1 g (*n* = 1) Cefuroxime 1.5 g (*n* = 1) Meropenem 1000 mg (*n* = 1)
*Pseudomonas aeruginosa*	Ceftazidime 1 g (*n* = 2) Ceftriaxone 1 g (*n* = 1) Colistin 1 MUI (*n* = 1) Linezolid 2 mg/mL (*n* = 1) Meropenem 1000 mg (*n* = 3) Piperacillin/Tazobactam 2 g/0.25 g (*n* = 2) Vancomycin 1 g (*n* = 2)	Amikacin 500 mg (*n* = 1) Cefazolin 1 g (*n* = 1) Ceftriaxone 1 g (*n* = 2) Clindamycin 300 mg/2 mL (*n* = 1) Colistin 1 MUI (*n* = 2) Fosfomycin 3 g p.o. (*n* = 1) Meropenem 1000 mg (*n* = 3) Metronidazole 250 mg (*n* = 2) Metronidazole 5 g/200 mL (*n* = 2) Tigecycline 50 mg (*n* = 2) Vancomycin 1 g (*n* = 1)
*Staphylococcus aureus*	Amoxicillin/Clavulanic acid 1000/200 mg (*n* = 2) Ceftriaxone 1 g (*n* = 2) Clindamycin 300 mg/2 mL (*n* = 1) Oxacillin 1000 mg (*n* = 2) Piperacillin/Tazobactam 2 g/0.25 g (*n* = 1) Trimethoprim/Sulfamethoxazole 400/80 mg (*n* = 1)	Amoxicillin/Clavulanic acid 1000/200 mg (*n* = 1) Ceftriaxone 1 g (*n* = 2) Clindamycin 300 mg/2 mL (*n* = 1) Colistin 1 MUI (*n* = 1) Erythromycin 200 mg (*n* = 1) Meropenem 1000 mg (*n* = 1) Metronidazole 5 g/200 mL (*n* = 1) Penicillin G potassium 1 MUI (*n* = 1) Vancomycin 1 g (*n* = 2)
*Staphylococcus epidermidis*	Amoxicillin/Clavulanic acid 1000/200 mg (*n* = 1) Meropenem 1000 mg (*n* = 1) Vancomycin 1 g (*n* = 1)	Ceftriaxone 1 g (*n* = 2) Cefuroxim 1.5 g (*n* = 1) Meropenem 1000 mg (*n* = 1) Vancomycin 1 g (*n* = 1)
*Staphylococcus hemolyticus*	Meropenem 1000 mg (*n* = 1) Metronidazole 5 g/200 mL (*n* = 1) Teicoplanin 400 mg (*n* = 1) Trimethoprim/Sulfamethoxazole 400/80 mg (*n* = 1) Vancomycin 1 g (*n* = 1)	Ceftriaxone 1 g (*n* = 4) Colistin 1 MUI (*n* = 1) Linezolid 2 mg/mL (*n* = 1) Meropenem 1000 mg (*n* = 2) Tigecycline 50 mg (*n* = 1) Vancomycin 1 g (*n* = 2)

## Data Availability

Data unavailable due to ethical restrictions.

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
