# Peer review of "Evaluation of the Bacterial Infections and Antibiotic Prescribing Practices in the Intensive Care Unit of a Clinical Hospital in Romania"

_antibiotics, 2025, doi:10.3390/antibiotics14010064_

Round 1

Reviewer 1 Report

Comments and Suggestions for Authors

To authors; 

Your study subject is very important, but you need to improve. Healthcare-associated infections are acquired while patients are receiving healthcare. At least 48 hours after hospitalization or 30 days after discharge admission (up to one year if there is an implant). What are your inclusion criteria for the 125 patients included in your study? "Your retrospective study was conducted in the ICU ward in a clinical hospital from Cluj-Napoca between 1st of February 2020 and 31st of December 2020 in Cluj-Napoca, Romania." This means an 11-month period. Your tables are not understandable. Numbers given on bacteria are inconsistent. For example; According to Table 1, there were 82 growth! I do not understand what 71 means! Samples were collected from 43 patients of 125. In this line; 71+24+21+9=125. But, you say the number of growing bacteria (1, 2, >2 ), and this number equals 51.   

 Table S1 is unnecessary detail. It makes the article longer like a book! AST results for bacteria were not provided.

Since "the intensive care unit" is mentioned the title of the article, more emphasis should be placed on this part.

I have some suggestions and corrections in the attached file.

Best regards

Comments on the Quality of English Language

A native English speaker should proofread it. Some sentences are difficult to understand and it is not appropriate to reuse some words in the same sentence.

Author Response

We thank the reviewer for the valuable suggestions that will help us improve the manuscript's quality.

Q1. Your study subject is vital, but you need to improve. Healthcare-associated infections are acquired while patients are receiving healthcare. At least 48 hours after hospitalization or 30 days after discharge admission (up to one year if there is an implant).

What are your inclusion criteria for the 125 patients included in your study?

Response: We had better underline the inclusion criteria in the revised manuscript.

Q2. "Your retrospective study was conducted in the ICU ward in a clinical hospital from Cluj-Napoca between 1st of February 2020 and 31st of December 2020 in Cluj-Napoca, Romania." This means an 11-month period.

Response: Yes, our cohort reflects the 11 months.

Q3. Your tables are not understandable. Numbers given on bacteria are inconsistent. For example; According to Table 1, there were 82 growth! I do not understand what 71 means! Samples were collected from 43 patients of 125. In this line; 71+24+21+9=125. But, you say the number of growing bacteria (1, 2, >2 ), and this number equals 51.   

Response: As presented in Table 1, we have 43 patients from whom no biological samples were collected. The possible value for growing bacteria also includes 0, so no bacterium growth. More than one bacterium was identified in some samples. The sum of data presented in the table will not lead to the total number of evaluated patients because a patient could have more than one biological sample and more than one bacterium could be identified in a sample.

Q4. Table S1 is unnecessary detail. It makes the article longer like a book!

Response: We agree, thank you for your suggestion. We excluded the supplementary table from the revised manuscript.

Q4. AST results for bacteria were not provided.

Response: We reported in Table 1 the number of patients with at least one R/I/S.

Q5. Since "the intensive care unit" is mentioned in the title of the article, more emphasis should be placed on this part.

Response: Thank you for your suggestion. We updated the Introduction to include more information about the ICU.

Q6. I have some suggestions and corrections in the attached file.

Response: Thank you for your support and suggestions. Your suggestions are valuable, and we modified the manuscript accordingly.

Reviewer 2 Report

Comments and Suggestions for Authors

The manuscript by Cecilia Cristea et al on bacterial infections and antibiotic prescribing practices in the intensive care unit of a clinical hospital in Cluj, Romania presents an in-depth analysis of the subject. The paper should be published after in-house cosmetic editing. Large data analysis as used here is clearly a subject of impact in clinical treatments and the large cohort size ensures the quality of the work.

The manuscript is ell structured and the English good. The presentation is clear and understandable.

The results, discussion and conclusions are well presented and for myself represent a clear in this area.

Publish and support.

Author Response

Reviewer 2 :

The manuscript by Cecilia Cristea et al on bacterial infections and antibiotic prescribing practices in the intensive care unit of a clinical hospital in Cluj, Romania presents an in-depth analysis of the subject. The paper should be published after in-house cosmetic editing. Large data analysis as used here is clearly a subject of impact in clinical treatments and the large cohort size ensures the quality of the work.

The manuscript is well structured and the English good. The presentation is clear and understandable.

The results, discussion and conclusions are well presented and for myself represent a clear in this area.

Publish and support.

Response: We thank the reviewer for the kind words.

Reviewer 3 Report

Comments and Suggestions for Authors

I have read with interest the manuscript submitted by Szabo et al, since AMR represents a global concearn.

I have a few commetns to be addressed in order to improve the quality of the manuscript:

-rows 20-21 - higher than what?

- I recommend using square brackets for references.

- rows 44-51 - I suggest adding information also from this study: 

Self-reported antibiotic stewardship and infection control measures from 57 intensive care units: An international ID-IRI survey

by Prof. Hakan Erdem et al., and also some more information about AMR in Romania, since the resistance rates are alarmingly higher than those from western Europe, especially among strains of Enterobacterales.

- clear inclusion/exclusion criteria should be stated;

- the terms dead/alive seem a bit raw. Consider switching to more elegant terms such as deceased, mortality, survived, favorable outcome, etc.

- any statistical comparisons for Table IV?

- Could the authors include some information about the resistance profile? When talking about antibiotic prescribing practices, this information is of paramount importance.

- what were the risk factors associated with mortality (univaritate/multivariate analisys)?

- ICU-acquired tract infections?

- the word multiresistant should be replaced with MDR (multudrug-resistant)

I highly suggest the revision of this manuscript by an Infectious Diseases specialist.

- all Latin bacterial names should be italicized in the discussion section as well.

There is information in the discussion section that was not included in the results! (rows 350-365)

- Results for susceptibility testing are available only after two days - why 2 days? Depending on the method, it can variate a lot

- The prescription date of antibiotics is not always identical to the first day of treatment - how many patients in your studied group had inappropriate empirical treatment? Did this influenced the outcome?

- rows 372-382 - these limitations can be easily overcomed if the patients charts are analyzed, not relying only the elecgtronic data.

- the conclusion section is too long. It should be focused only on the main findings of this study. No references should be included in this section.

- the reference list is scarce and not edited according to the mdpi pattern

Author Response

We thank the reviewer for the valuable suggestions that will help us to improve the quality of the manuscript.

I have read with interest the manuscript submitted by Szabo et al, since AMR represents a global concearn.

Response: Thank you for your constructive feedback.

I have a few comments to be addressed in order to improve the quality of the manuscript:

Q1. rows 20-21 - higher than what?

Response: Higher than the group of discharged patients. The phrase was modified for more clarity.

Q2. I recommend using square brackets for references.

Response: The revised manuscript fully respects the journal template.

Q3. rows 44-51 - I suggest adding information also from this study: Self-reported antibiotic stewardship and infection control measures from 57 intensive care units: An international ID-IRI survey by Prof. Hakan Erdem et al.,

Response: Thank you for your suggestion. Information related to specified reference is included in the revised manuscript in Introduction section.

Q4. also some more information about AMR in Romania, since the resistance rates are alarmingly higher than those from western Europe, especially among strains of Enterobacterales.

Response: Information regarding Antibiotic Consumption, Microbial Resistance and Healthcare-Associated Infections in Romania – 2020 was included in the Discussion section. There are not too many information about Romania, however we added the most recent data from Romania.

Q5.- clear inclusion/exclusion criteria should be stated

Response: The inclusion criteria were better underlined.

Q6.  the terms dead/alive seem a bit raw. Consider switching to more elegant terms such as deceased, mortality, survived, favorable outcome, etc.

Response: Thank you for the suggestion. The manuscript was modified according to the suggestion.

Q7. any statistical comparisons for Table IV?

Response: In the revised manuscript, we reported for the top 5 most used antibiotics the comparison between groups.

Q8. Could the authors include some information about the resistance profile? When talking about antibiotic prescribing practices, this information is of paramount importance.

Response: We included a new section in the results regarding the resistance.

Q9. what were the risk factors associated with mortality (univaritate/multivariate analisys)?

Response: Using the available data, we were not able to identify any risk factors for mortality neither in univariate nor in multivariate analysis. No changes were made in the revised manuscript regarding this comment.

Q10.- ICU-acquired tract infections?

Response: This phrase was corrected.

Q11. the word multiresistant should be replaced with MDR (multudrug-resistant)

Response: Thank you for the suggestion. The manuscript was modified according to the suggestion.

Q12. I highly suggest the revision of this manuscript by an Infectious Diseases specialist.

Response: The revised manuscript was carefully verified by the coauthors which are Infectious Diseases medical doctors.

Q13. all Latin bacterial names should be italicized in the discussion section as well.

Response: Thank you for the suggestion. The manuscript was modified according to the suggestion.

Q14. There is information in the discussion section that was not included in the results! (rows 350-365)

Response: Thank you for your observation. We moved the indicated paragraph from Discussion to a new section created in the Results.

Q15. Results for susceptibility testing are available only after two days - why 2 days? Depending on the method, it can variate a lot

Response: We included in the methods section the protocol used in the hospital.

Q16. The prescription date of antibiotics is not always identical to the first day of treatment - how many patients in your studied group had inappropriate empirical treatment? Did this influenced the outcome?

Response: The prescribing program is a basic one, it often gets blocked, so the antibiotic is dispensed from the pharmacy, administered to the patient, but only the second or third day is prescribed electronically. At the same time, there are antibiotics on the wards that can be administered to patients but are only registered into the electronic system later (sometimes the second day) due to the high workload or insufficient nursing staff. (Nurses are responsible for drug prescribing in the electronic system).

Q17. Rows 372-382 - these limitations can be easily overcomed if the patients charts are analyzed, not relying only the elecgtronic data.

Response: After patients' discharge or death, patinet charts are archived, and access to these documents requires approvals, so we considered electronic data analysis.

Q18. the conclusion section is too long. It should be focused only on the main findings of this study. No references should be included in this section.

Response: We modified the title to Conclusions and future trends.

Q19. the reference list is scarce and not edited according to the mdpi pattern

Response: In the revised manuscript, the formatting of the MDPI was fully respected.

Round 2

Reviewer 1 Report

Comments and Suggestions for Authors

I have some suggestions in the attached file.

Author Response

Thank you for your useful suggestions, the suggested corrections were marked in yellow in the main manuscript.

Reviewer 3 Report

Comments and Suggestions for Authors

I appreciate all the author's efforts in addressing my comments.

The quality of the manuscript has improved. However, I still have some remarks to be addressed:

- once an abbreviation is defined, further mentions should use only the acronym.

-  Healthcare–associated infections is  are associated

- most frequent bacteria were was A. baumannii 

- Acinetobacter baumannii, Klebsiella pneumoniae, Pseudomonas aeruginosa was were most....

Multiple typos and misuse of words were identified. I highly recommend the manuscript be double-checked by an English-speaking person. The newly inserted paragraphs are often hard to understand.

in the discussion section, no reference to tables/figures should be included

- all Latin bacterial names should be italicized. Moreover, it is indicated that the first use in the text of a Latin bacterial name should be with the complete form, and all the rest with the abbreviated one (ex. Escherichia coli -> E. coli)

- not enough data on AMR locally was inserted in the revised manuscript. Here are some examples of recent data in this regard 10.3390/antibiotics12020324, 10.3390/antibiotics11050548, 10.1007/s10096-021-04288-1, 10.3390/antibiotics10050523

Best regards,

Comments on the Quality of English Language

Extensive editing is required, mostly (but not limited to) the newly inserted paragraphs.

Author Response

I appreciate all the author's efforts in addressing my comments.

The quality of the manuscript has improved. However, I still have some remarks to be addressed:

We thank the reviewers for the useful suggestions that improved the quality of our manuscript.

- once an abbreviation is defined, further mentions should use only the acronym.

Answer: Revised, except when it is the first in the sentence.

-  Healthcare–associated infections is  are associated

Answer: It was revised.

- most frequent bacteria were was A. baumannii

Answer: it was revised.

- Acinetobacter baumannii, Klebsiella pneumoniae, Pseudomonas aeruginosa was were most....

Answer: It was revised.

Multiple typos and misuse of words were identified. I highly recommend the manuscript be double-checked by an English-speaking person. The newly inserted paragraphs are often hard to understand.

Answer: the manuscript was carefully checked by a native English speaker and the typos were corrected and highlighted in yellow the manuscript.

in the discussion section, no reference to tables/figures should be included

Answer: It was revised. We refer to our tables/figures in the discussion section to support the story and to avoid the duplication of the result.

- all Latin bacterial names should be italicized. Moreover, it is indicated that the first use in the text of a Latin bacterial name should be with the complete form, and all the rest with the abbreviated one (ex. Escherichia coli -> E. coli)

Answer: All the Latin bacterial names were written in italics and were abbreviated in the manuscript after the first use.

- not enough data on AMR locally was inserted in the revised manuscript. Here are some examples of recent data in this regard 10.3390/antibiotics12020324, 10.3390/antibiotics11050548, 10.1007/s10096-021-04288-1, 10.3390/antibiotics10050523

Answer: Thank you for this helpful suggestion. The suggested references and a discussion about data on AMR locally were included on page 15 lines 314-330.